# Multifunctional Chitosan/Gold Nanoparticles Coatings for Biomedical Textiles

**DOI:** 10.3390/nano9081064

**Published:** 2019-07-24

**Authors:** Iris O. Silva, Rasiah Ladchumananandasivam, José Heriberto O. Nascimento, Késia Karina O.S. Silva, Fernando R. Oliveira, António P. Souto, Helena P. Felgueiras, Andrea Zille

**Affiliations:** 1Department of Mechanical Engineering, Federal University of Rio Grande do Norte, Natal 59064-741, Brazil; 2Department of Textile Engineering, Federal University of Rio Grande do Norte, Natal 59064-741, Brazil; 3Department of Textile Engineering, Federal University of Santa Catarina (UFSC), Campus Blumenau, Blumenau 89036-002, Brazil; 4Centre for Textile Science and Technology, Department of Textile Engineering, University of Minho, 4800-058 Guimarães, Portugal

**Keywords:** multifunctional textiles, biodegradable soybean fibres, chitosan, gold nanoparticles, UV-light protection, antimicrobial

## Abstract

Gold nanoparticles (AuNPs), chemically synthesized by citrate reduction, were for the first time immobilized onto chitosan-treated soybean knitted fabric via exhaustion method. AuNPs were successfully produced in the form of highly spherical, moderated polydisperse, stable structures. Their average size was estimated at ≈35 nm. Successful immobilization of chitosan and AuNPs were confirmed by alterations in the fabric’s spectrophotometric reflectance spectrum and by detection of nitrogen and gold, non-conjugated C=O stretching vibrations of carbonyl functional groups and residual N-acetyl groups characteristic bands by X-ray photoelectron spectroscopy (XPS) and Fourier-Transform Infrared Spectroscopy (FTIR) analysis. XPS analysis confirms the strong binding of AuNPs on the chitosan matrix. The fabrics’ thermal stability increased with the introduction of both chitosan and AuNPs. Coated fabrics revealed an ultraviolet protection factor (UPF) of +50, which established their effectiveness in ultraviolet (UV) radiation shielding. They were also found to resist up to 5 washing cycles with low loss of immobilized AuNPs. Compared with AuNPs or chitosan alone, the combined functionalized coating on soy fabrics demonstrated an improved antimicrobial effect by reducing *Staphylococcus aureus* adhesion (99.94%) and *Escherichia coli* (96.26%). Overall, the engineered fabrics were confirmed as multifunctional, displaying attractive optical properties, UV-light protection and important antimicrobial features, that increase their interest for potential biomedical applications.

## 1. Introduction

For many years, the textile industry has been responsible for the development of functional fabrics with improved comfort. The growing demands for sophisticated and multifunctional textiles boosted the integration of multidisciplinary and nanotechnology approaches within the textile traditional sector. Indeed, a wide range of nanosized materials endowed with antimicrobial properties, high surface energy, small size and large surface area, and high affinity towards fabrics, have been employed in textile finishing processes to improve their durability, protection and health benefits, without compromising fabrics’ breathability and handfeel properties [1,2].

Immobilization of noble metal nanoparticles onto textile fibres and fabrics pioneered innovative textile functional properties, including Ultraviolet (UV)-light protection and antimicrobial action [3,4]. These nanoparticles have also been responsible for generating bright colours by taking advantage of their particular localized surface plasmon resonance (spr) optical properties [5]. Gold nanoparticles (AuNPs) are the most stable metal nanoparticles (NPs), resisting to oxidation and chemical attacks [6]. They are inert and relatively low cytotoxic, exhibiting fascinating electronic, magnetic, and optical properties that can be tuned according to the particles’ size, shape, surface chemistry or aggregation state, making them very promising in the fields of biosensors, catalysis, nanophotonics, biotechnology and biomedicine [7]. For biomedical applications, AuNPs are frequently engineered as water-soluble and aggregation-resistant colloids, capable of enduring in vivo environments [8]. Monodispersed, small sized AuNPs are recommended to enable multivalency contact surface with bacteria, changing their metabolite pathways and release mechanisms, and thus enhancing their antimicrobial action against both Gram-positive and Gram-negative bacteria [9,10].

There have been many reports on the functionalization of fibrous materials with AuNPs [11]. However, there are very few reports on the antimicrobial direct effect of AuNPs immobilized on fibrous material. Most of the researches reports on antimicrobial effect of liquid suspensions [12,13]. The few previous works on fibrous materials are barely related to our work because they mostly assessed chemically-functionalized AuNPs or non-spherical (nor pseudo-spherical) shaped AuNPs [4,5]. For instance, Dong et al. established electrostatic interactions, as the driven mechanisms responsible for high surface coverage of natural cellulose fibrous constructs (cationic) with AuNPs (anionic). They also observed that high surface coverage could be obtained from low concentration precursor solutions [14]. On its turn, Radic et al. determined ambient air plasma treatments on polypropylene nonwoven fabrics to enhance deposition of AuNPs. The resultant surfaces were seen to possess highly improved wetting and sorption properties, and to exhibit antibacterial activity against *Staphylococcus aureus* and *Escherichia coli* [15]. Johnston et al. produced a range of coloured wool fibres by chemically binding AuNPs of different sizes and colours to wool keratin, and generated stable colourfast colourants that did not fade compared to traditional organic dyes [6]. Tang et al. coloured silk fabrics in red and brown using AuNPs with particular localized spr properties and demonstrated the resultant fabrics to possess good light fastness, excellent UV-light protection, enhanced thermal conductivity and strong antibacterial activity [5]. Later, similar multifunctional performances were attained, while synthesizing AuNPs in situ on bamboo pulp fabrics [16]. Most of the studies reporting on antimicrobial Chitosan-AuNPs nanocomposites focus on membranes and not on fibrous materials. In addition, the Chitosan-AuNPs in solid films were only analysed in terms of physicochemical properties and antimicrobial activity for the first time in 2015 [17]. The antibacterial activity is frequently assessed by the American Association of Textile Chemists and Colorists (AATCC) 147 parallel streak method that works very well only for very thin woven fabrics or by the AATCC 100 assessment of antibacterial finishes on textiles that it is known for its vague success criteria, especially if the fabric does not readily absorb liquids; while our work is based on the more realistic Shake Flask Test Method, which is designed to evaluate the resistance of non-leaching (or also leaching) antimicrobial treated specimens to the growth of microbes under dynamic contact conditions [16]. Moreover, to the authors’ knowledge, immobilisation of AuNPs onto soybean fabrics has yet to be explored. Soybean protein fibres have become particularly prominent because of their abundance, renewable, biodegradable and cost-effective properties compared to cotton, wool or silk [18]. They exhibit low specific gravity, high elongation, enhanced moisture absorption, improved permeability and heat-retaining performances, good acidic and alkali resistance, and greater tensile strength than other natural-origin fibres (i.e., cotton, wool and silk) [19]. The successful manufacture of soybean fibres is a breakthrough that responds to the main weaknesses of other regenerated fibres, the environmental impact (pollution and use of non-renewable resources). Indeed, both soybean fibres production and chemical treatment, which is known to enhance macromolecules fixation, are eco-friendly [20,21]. One major drawback to the use of soybean fabrics in biomedicine remains the inability to fight microorganisms invasion and colonization [22]. The present work addresses this issue by incorporating AuNPs, chemically synthesized by citrate reduction, into soybean knitted fabrics pre-treated with chitosan. A large number of methods, including coupling/grafting of low or high molecular weight compounds, have been employed to modify the surface of natural and synthetic fibres and improve NPs’ binding [23]. However, the immobilization of AuNPs into fibrous materials is an unexplored subject especially their antimicrobial action mode, which remains unclear [17]. Chitosan is a nontoxic, biodegradable macromolecule with emerging applications in drug delivery and tissue engineering. Its solubility in slightly acidic solutions allows chitosan to be co-processed with other biomolecules, as its primary amines can be chemically protonated, resulting in a positively charged polysaccharide. Its cationic nature endows chitosan with the ability to form complexes with a variety of polyanions, becoming the perfect binding agent for NPs [24].

The final AuNPs functionalized soybean knitted fabrics were characterized in terms of NPs size and distribution, chemical composition by FTIR and XPS analyses, thermal properties and stability by DSC and TGA, respectively, UV-light protection, colour fastness and bactericidal effect.

## 2. Experimental

### 2.1. Materials

Soybean protein-based knitted rib fabric [prepared with polyvinyl alcohol (PVA)] of 95 g m^−2^ of weight per unit area was used in this study. Tetrachloroauric acid (HAuCl_4_) with 99.5% purity was purchased from PLAT LAB (Guarulhos, SP, 07170-560, Brazil), sodium citrate dihydrated (Na_3_C_6_H_5_O_7_·2H_2_O) was obtained from Êxodo Científica (Sumaré, SP, 13175-695, Brazil), and chitosan with deacetylation grade of 90.57% and viscosity of 86 mPa s was acquired from the Golden Shell Biochemical Co., Ltd. (Zhejiang, TE86-576-87259169, China). The remainder analytical grade reagents were obtained from Sigma-Aldrich (Jurubatuba, São Paulo, SP, 04696-010, Brazil) and used without further purification.

### 2.2. AuNPs Synthesis

AuNPs were chemically synthesized by a modified stepwise method of the conventional citrate reduction technique. Briefly, 0.01% *w*/*v* of HAuCl_4_ was prepared in distilled water (dH_2_O) and heated at 90 °C under constant stirring. To this solution, 2 mL of 1% *w*/*v* Na_3_C_6_H_5_O_7_.2H_2_O in dH_2_O were added dropwise. After 3 min of stirring, the solution started changing colour, from grey to blue and then purple until finally reaching the colour red, indicating the AuNPs were formed. The solution was left under constant stirring for another 30 min. In the end, it was removed from the heater and stirred until reaching room temperature (RT).

### 2.3. Chitosan Treatment

Chitosan solution was prepared at 2% *w*/*v* in 50 mL of dH_2_O and 0.5 mL of acetic acid (CH_3_COOH). The solution was stirred for 2 h at RT and twice filtered. Dilutions at 5, 7, 10 and 20% *v/v* in the same solvents were prepared. 1.5 g samples of soybean knitted fabric were combined in a proportion of 1:65 with each solution. Bare fabric samples and samples immersed in chitosan were used as control. Using the ALT-B TOUCH 35 equipment from Mathis Company (8156 Oberhasli, Switzerland), all samples were submitted to exhaustion for 30 min at 40 rpm, temperature of 70 °C, pressure of 4 bar and auto-reversion of 50 s. In the end, the samples were collected and dried at 70 °C for 1 h.

### 2.4. AuNPs Functionalization

1.5 g samples of soybean knitted fabric treated with 20% chitosan dilution were immersed in 97.5 mL baths of dH_2_O and CH_3_COOH (10/5 *v/v*) containing a pure colloidal suspension of AuNPs. The temperature and time of the exhaustion method were adjusted to 70 °C and 30 min. Finally, the samples were washed in dH_2_O for 30 s, dried for 3 min at 100 °C, and cured for 2 min at 120 °C.

### 2.5. Scanning Electron Microscopy (SEM) and Energy Dispersive X-ray Spectroscopy (EDS)

Morphological analyses of the fabrics before and after AuNPs functionalization were carried out with an Ultra-high-resolution Field Emission Gun SEM (FEG-SEM), NOVA 200 Nano SEM, FEI Company, Hillsboro, OR, USA. Secondary electron images were acquired with an acceleration voltage of 5 kV. Backscattering electron images were realized with an acceleration voltage of 15 kV. Samples were covered with a thin film of Au-Pd (80–20 wt%) in a high-resolution sputter coater, 208HR (Cressington Company, Watford WD19 4BX, United Kingdom) coupled to an MTM-20 Cressington High Resolution Thickness Controller. Atomic compositions of the membrane were examined by EDS using an EDAX Si(Li) detector at acceleration voltage of 5 kV.

### 2.6. Dynamic Light Scattering (DLS) and Zeta Potential Measurements

The AuNPs size distribution, polydispersity index and zeta potential were measured by dynamic light scattering (DLS) and electrophoretic light scattering (ELS) using a Zeta Sizer-Nano (Malvern Panalytical Ltd, Malvern, WR14 1XZ, United Kingdom). Data was collected after 30 scans at a constant temperature of 25 ± 1 °C. Zeta potentials were measured in solution at a moderate electrolytic concentration. Each value was obtained by averaging measurements of ten samples.

### 2.7. UV-Visible Spectroscopy

UV-Vis spectroscopy (Agilent Cary-50 spectrophotometer, Agilent Technologies, Santa Clara, CA, USA) was used to evaluate the dyeing uptake efficiency of the colloidal Au dispersion on the fabric, in the range between 300 and 800 nm.

### 2.8. Attenuated Total Reflectance-Fourier-Transform Infrared Spectroscopy (ATR-FTIR)

A Nicolet Avatar 360 FTIR spectrophotometer (Waltham, MA, USA) with an ATR accessory was used to record the ATR-FTIR spectra of the fabrics, performing 45 scans at a spectral resolution of 4 cm^−1^ over the range 650–4000 cm^−1^. All measurements were performed in triplicate.

### 2.9. X-ray Photoelectron Spectroscopy (XPS)

XPS analyses were performed using a Kratos AXIS Ultra HSA, with VISION software for data acquisition (Kratos Analytical Ltd, Manchester, United Kingdom) and CASAXPS software (Casa Software ltd., Teignmouth, Devon, TQ14 8NE, United Kindom, version 2.3.15) for data analysis. XPS analysis was carried out with a monochromatic Al Kα X-ray source (1486.7 eV), operating at 15 kV (150 W) in FAT mode (Fixed Analyzer Transmission), with pass energy of 40 eV for regions ROI and 80 eV for survey. Data was collected with pressure lower than 1×10^6^ Pa and using a charge neutralization system. Spectra were charge corrected to give the adventitious C1s spectral component (C–C, C–H) a binding energy of 285 eV. High-resolution spectra were collected using an analysis area of ≈1 mm^2^. The peaks were constrained to have equal FWHM to the main peak. This process has an associated error of ±0.1 eV. Spectra were analysed for elemental composition using CasaXPS software. Deconvolution into sub-peaks was performed by least-squares peak analysis software, XPSPEAK (Free Software written by Prof. Raymond W.M. Kwok, Chinese University of Hong Kong, version 4.1), using the Gaussian/Lorenzian sum function and Shirley-type or linear background subtraction. No tailing function was considered in the peak fitting procedure. The components of the various spectra were mainly modeled as symmetrical Gaussian peaks unless a certain degree of Lorentzian shape was necessary for the best fit. The best mixture of Gaussian–Lorentzian components was defined based on the instrument and resolution (pass energy) settings used as well as the natural line width of the specific core hole. The reduced chi-square is the sum of the squares of the difference between the experimental spectrum and the fitted envelope at each point over the peak region of interest, divided by the variance. A reduced chi-square value of less than or equal to 2 is typically regarded as indicative of an acceptable peak-fit. Photoelectron peak area (counts) two times larger than the background noise, not intensities, was used for quantitative analysis.

### 2.10. Thermal Gravimetric Analysis (TGA)

TGA was performed on a STA 449 F3 from NETZSCH Q500 using a platinum pan (Netczsch Brazil, 04576.060, São Paulo, SP, Brazil). The TGA trace was obtained in the range of 30–600 °C under nitrogen atmosphere, flow rate of 20 mL min^−1^ and temperature rise of 20 °C min^−1^. Results were plotted as percentage of mass loss vs. temperature.

### 2.11. Differential Scanning Calorimeter (DSC) Analysis

DSC was carried on a Power Compensation Diamond DSC (Perkin Elmer, Waltham, MA, USA) with an Intracooler ILP, based on the standards ISO 11357-1:1997, ISO 11357-2:1999 and ISO 11357-3:1999. Samples were dried at 60 °C for 1 h and placed in an aluminium sample pan before testing. The analysis was carried out in nitrogen atmosphere with a flow rate of 20 mL/min and heating rate of 20 °C/min. The thermogram was obtained in the range of 30 °C to 350 °C. Graph was plotted with heat flow vs. temperature.

### 2.12. Diffuse Reflectance Spectroscopy

The reflectance of AuNPs functionalized fabrics treated with 5, 7, 10 and 20% chitosan dilutions were evaluated using a Konica Minolta CM-2600d (Konica Minolta, Chiyoda, Japan) diffuse fibre optic reflectance spectrophotometer, in the range between 400 and 700 nm, at standard illuminant D65 (LAV/Spec. Incl., d/8, D65/10°). Five areas on each sample were measured in various positions, and the results presented as average values with up to 1% variation. All measurements were performed in triplicate. The responses analysed were the colour characteristics: K/S, L*, a*, b*. K/S is the colour strength calculated using Kubelka-Munk’s equation (K/S = (1-R)^2^/2*R, where R is the reflectance). L*, a*, and b* are the coordinates of the colour in the cylindrical colour space, based on the theory that colour is perceived by black-white (L*, lightness), red-green (a*), and yellow-blue (b*) sensations. The lightness, L*, represents the darkest black at L* = 0 and the brightest white at L* = 100. The colour channels, a* and b*, will represent true neutral gray values at a* = 0 and b* = 0. The red/green opponent colours are represented along the a* axis, with green at negative a* values and red at positive a* values. The yellow/blue opponent colours are represented along the b* axis, with blue at negative b* values and yellow at positive b* values.

### 2.13. UV-Light Protection

The ultraviolet transmittance UVB (280–315 nm) and UVA (315–400 nm) and the ultraviolet protection factor (UPF) of fabric samples were measured by Camspec M350 SPF Spectrophotometer (SDL ATLAS, Rock Hill, SC, USA) according to AATCC Test Method 183–2004. The samples were placed in a spectrophotometer and scanned between 290 and 400 nm, with a 5 nm interval at RT. Data was provided in the form of average of five consecutive tests at different positions.

### 2.14. Colour Fastness

The resistance of the functionalized AuNPs on a 20% chitosan-treated fabric to fading was evaluated using the standard AATCC 61–2006. Briefly, 5 samples of 5 × 10 cm^2^ were immersed in a 150 mL water bath containing 0.15% spm of coconut soap and submitted to 40 rpm at 40 ± 2 °C for 1 h, using the ALT-B TOUCH equipment (Werner Mathis AG, 8156 Oberhasli, Switzerland). Five washing cycles were conducted. Results were analysed qualitatively against a grey scale from 1 (low resistance) to 5 (excellent resistance).

### 2.15. Antimicrobial Testing

The antibacterial efficacy of the AuNPs functionalized soybean knitted fabrics was assessed quantitatively following the standard shake flask method (ASTM-E2149-01), described previously [25]. Both Gram-positive and Gram-negative bacteria were used, respectively *Staphylococus aureus* (*S. aureus*) and *Escherichia coli* (*E. coli*). The test was performed aseptically to ensure the absence of any contamination. Bacteria inoculum were prepared from a single colony and incubated overnight in tryptic soy broth (TSB, Merck) at 37 °C and 120 rpm. Each test was carried out using an initial concentration of 1.5–3.0 × 10^7^ CFUs/mL in phosphate buffer saline solution (PBS pH 7.4). AuNPs functionalized fabric samples of 0.05 g weight were incubated in 5 mL of bacteria suspension at 37 °C and 100 rpm. After 0 h (before contact with sample) and 24 h of culture, the bacteria were serially diluted, cultured onto tryptic soy agar (TSA, Merck KGaA, Darmstadt, Germany) plates, and further incubated for another 24 h. Quantitative results were obtained by counting the colonies of surviving bacteria on the agar plates. Antimicrobial activity was reported qualitatively in terms of percentage of bacteria reduction calculated as the ratio between the number of surviving bacteria colonies present on the TSA plates, before and after contact with fabric. All antibacterial tests were conducted in triplicate and represent in the form of mean ± SD.

## 3. Results and Discussion

### 3.1. Fabric Reflectance

The most effective chitosan concentration for AuNPs binding was selected between solutions at 5, 7, 10 and 20% dilution. Binding efficiency was determined by monitoring the percentage of fabric reflectance between 400 and 700 nm (Figure 1). It was seen that the higher the chitosan concentration, the higher the amount immobilized and consequently the lower the reflectance. Subsequent testing was conducted using 20% chitosan concentration, which allowed for an even coating above the fabric and the highest cationization effect. It was observed that at higher chitosan concentration, the coating onto the fabric was not uniform and chitosan agglomerates started to appear to compromise an efficient AuNPs deposition.

### 3.2. AuNPs Characterization

The size and colloidal dispersion of the AuNPs were evaluated via UV-visible spectroscopy and DLS. Reduction of AuCl_4_^−^ ions (precursor) into AuNPs was confirmed visually by changes in the solution colour, going from a pale yellow to a ruby red (Appendix A). It is well established that AuNPs reduction results in a ruby red colour in aqueous solutions due to the excitation of the spr vibrations, located between 520 and 540 nm (Figure 2A) [26]. This adsorption band is also an indicator of the presence of monodispersed NPs in the colloidal solution, and consequent absence of aggregates [27]. At high temperatures such as 90 °C, most gold ions form nuclei very quickly acquiring a small, spherical-like shape. Since the reaction rate is very high, consuming all the metallic specimens in the system, the secondary growth of the particles stops preventing more complex morphologies from being formed (i.e., triangles, pentagons, and hexagons) [28].

DLS data estimates the average size of the AuNPs at 34.6 ± 0.5 nm and the average polydispersity index at 0.3, confirming a moderate polydispersivity [29]. Figure 2B confirms this statement and establishes the NPs morphology as highly spherical. The zeta potential (ζ) of the AuNPs, which provides useful information about the NPs charge, potential stability of the colloid and ability to interact with other molecules, was determined at −37.4 ± 5.7 mV and, thus, considered relatively stable [30]. It also suggests that the relative stability of the AuNPs in the dispersion to result from the electric repulsion of the negative charge of the citrate ions adsorbed onto the NPs surface. Citrate-reduced AuNPs are negatively charged, as citrate ions are attracted towards the surface of the Au NPs, conferring some degree of stability to the NPs [31]. ζ depends of the ionic strength and the dielectric constant of the media. An increase in the dielectric constant is translated into an increase of the effective potential of the NPs in the shear plane [32].

### 3.3. AuNPs Functionalization

Soybean-knitted fabrics were functionalised with AuNPs via exhaustion method, using 20% (*v/v*) chitosan as a binding agent. Chitosan is a nontoxic, biodegradable macromolecule, soluble in slightly acidic solutions [23,33]. Its primary amines can be chemically protonated, serving as chelation sites for adsorption and binding of a number of metal polyanions. Chelation evenly disperses the citrate-reduced, negatively charged NPs binding throughout the polymer and prevents leaching [34]. Figure 3 shows the macroscopic and microscopic appearance of the soybean fabric before and after AuNPs functionalization. The darker colour (bluish-red) of the fabric (Figure 3b) is an indicative of the gold content and density at the surface, which is known to deepen as the amount of functionalized AuNPs increases [35]. Fabric functionalization was also confirmed by measuring the absorbance of the colloidal suspension before and after fabric immersion. The peak between 520 and 540 nm, characteristic of the AuNPs (Figure 2a), was no longer found after fabric immersion confirming the complete immobilization of the AuNPs (Appendix A). A large number of spherical aggregated AuNPs were observed distributed along the soybean fibres (Figure 3d). Aside from acting as a linker, chitosan also serves as a stabilizing agent when combined with the AuNPs. It promotes positive capping agent’s electrostatic repulsion between metal NPs, preventing particle aggregation [36]. Fabrics that were not submitted to a 20% chitosan pre-treatment were less densely functionalized and were predominantly populated by compact, large AuNPs agglomerates (Appendix A).

### 3.4. ATR-FTIR

Spectra of the soy fabrics untreated and coated with chitosan and/or AuNPs were collected (Figure 4). Due to the overlapping of bands, analogous spectra predominated and little differences were detected among bare and coated fabrics. A strong band between 3350 and 3200 cm^−1^ was observed on all surfaces and was associated with N–H and O–H stretching vibrations and with intramolecular hydrogen bonds. The adsorption bands found at 2916 and 2854 cm^−1^ were attributed to the C–H asymmetric and symmetric stretching vibrations of PVA, the base component of the soybean fabric [37,38]. The peak at 1740 cm^−1^ was only identified on the fabrics containing AuNPs, attesting to the success of the functionalisation process. It was assigned to the non-conjugated C=O stretching vibrations of the carbonyl functional groups, suggesting the binding of aldehydes/ketones with the AuNPs during citrate reduction or immobilization (with CH_3_COOH) [39]. In case of chitosan-treated fabrics, the presence of the characteristic residual *N*-acetyl groups was verified by identifiable bands at 1635 (C=O stretching of amide I), 1543 (N–H bending of the amide II) and 1327 cm^−1^ (C–N stretching of amide III) [40]. The absorption bands centred at 1635 and 1543 cm^−1^ were also associated to the stretching vibrations of the C=C and to the N–H bonds of amide I and amide II, which form the primary backbone of proteins, including the soybean proteins. The broad peak at 1404 cm^−1^ was assigned to CH_2_ bending vibrations, while the peak at 1080 cm^−1^ has been considered as the contribution of out-of-plane C-H bending, both belonging to the soy fabric [41].

### 3.5. XPS

The relative chemical composition (C, N, O and Au) and atomic ratios (O/C and N/C) of the soybean fabrics coated with AuNPs, chitosan and chitosan/AuNPs were analysed by XPS (Table 1). The presence of chitosan leads to significant differences in the atomic O/C and N/C ratios. The higher concentration of nitrogen in the chitosan-coated fabrics is a proof of chitosan adsorption since it is the only compound containing amino groups in a significant amount. These results, together with the three survey spectra performed in different regions of the fabrics (Appendix A), confirm the uniform deposition of the chitosan layer on the fibres surface [42]. In the presence of chitosan, the O/C ratio is three times and the N/C ratio two times higher than the soybean-based fabric control. The presence of AuNPs also showed an increase of the O/C ratio suggesting that AuNPs are able to oxidise the surface of the fabric. The content of gold immobilized onto the fabric pre-treated with chitosan is one order of magnitude higher than the AuNPs deposited in the untreated soybean fabric confirming the good affinity of chitosan for the immobilization of AuNPs [43]. The high resolution S(2p) spectra (data not shown) of the treated and untreated soybean fabrics, indicate that the low intensity signal (~0.15% atomic) centred at 168 eV could be related to the low levels of unoxidized (S^2+^) sulphur-containing amino acids at the soybean fibre surface and to the S^6+^ form of bound sulfamic acid [44].

The C1s spectra of the Soybean-based fabric can be deconvoluted in four peaks (Table 2 and Appendix A) centred at 285, 285.9, 287.1 and 288.5 eV attributed to the C-C, C–O/C–N, O–C-O/C=O, and O–C=O, respectively [45]. It was interesting to note that after the addition of chitosan or AuNPs there was a positive shift in the binding energies of the C–O/C–N and O–C–O/C=O peaks. The presence of chitosan significantly increases the relative percentage of binding energy area of these two peaks and, in addition, also displays a positive shift of the carboxylic component peak from 288.5 to 288.9 eV. The shift and increase of these peaks can be attributed to the fact that the chitosan has been successfully grafted onto the soybean-based fabric changing the chemical environment of the fibres. However, there is no noteworthy difference between the chitosan or chitosan/AuNPs coatings in term of deconvoluted area peaks in C1s envelope. The fabric with only deposited AuNPs shows a shift in C–O/C–N and O–C–O binding energies, however there is no changes in the carboxylic peak suggesting that the preferred site for the bond of NPs may be the single bonded oxygen moieties (hydroxyl) of the backbone PVA polymer or the amino groups of the amino/peptide functionalities of the soybean-based fibres [46].

The deconvoluted O1s spectrum in the control soybean-based fibres (Table 2 and Appendix A) can be fitted in two peaks assignable to oxygen atoms in C–OH/C–O–C (532.3 eV) and carboxylic O=C–C (534.5 eV) groups [47]. The presence of a small peak of carboxylic groups can be attribute to certain degree of oxidation of the fibres surface. In the case of the fibres treated with AuNPs due to the interaction of AuNPs deposited onto the fabric surface with the carboxylic groups of the oxidized fibres, only the peak at 532.3 eV can be deconvoluted. The high-resolution spectra of the O1s core-level obtained for chitosan coating was decomposed to three peaks with BE of 531.3, 532.8 and 534.5 eV. The O1s peaks at 531.3 eV and 532.8 eV are attributed to the oxygen signals of the N-C=O chemical bindings in *N*-acetylated-glucosamine units and to the C–OH/C–O–C of acetal and hemiacetal of chitosan structure, respectively [48]. After AuNPs addition to the chitosan-coated fabric, the peaks at 532.9 eV and 531.5 eV showed a slight shift toward higher binding energies compared with the fabric with only coated chitosan which can be attributed to the interaction of AuNPs with chitosan. There is no shift in the peak at 534.5 eV attributed to the carboxylic group suggesting that the preferred sites for the AuNPs bond are not the oxygen species but the amine ones [49].

The N1s deconvoluted spectra of untreated, chitosan and AuNPs separately treated soybean fabrics (Table 2 and Appendix A) showed the presence of peak intensity at 400.0 eV, which can be assigned both to the protein amino/peptide functionalities of soybean fibres and, when present, to the chitosan amino groups (NH_2_) [50]. The very low intensity of the untreated fabric of the N1s peak and the high intensifies of the C–C and C–O peaks in the C1s envelope clearly demonstrated the presence of the PVA as backbone polymer in the soybean-based fibre surface. Two peaks were observed in the N1s spectrum of chitosan/AuNPs treated fabric, one at 400 eV and the other at 402 eV corresponding to protonated amine groups of chitosan [51]. The presence of coordinated protonated amino groups confirm the predominance of amine sites in chitosan for metal sorption due to the charge transfer from amine sites to AuNPs. The electrostatic attraction between positively charged amino groups of the chitosan chains and the negatively charged AuNPs results in nanoparticles stabilization and coordination of the protonated amino group of chitosan on the nanoparticle surface due to the charge transfer on nitrogen resulting in strong bonds. Moreover, the peak at 402 eV for N1s that only appeared in the sample with both AuNPs and chitosan confirms that the distribution of the AuNPs/chitosan interactions is not the same on the surface and in the bulk of the nanoparticles [52,53].

The deconvolution of the Au4f envelope in the AuNPs containing coatings was also performed to confirming the chemical behaviours of the metal nanoparticles in the presence or absence of chitosan (Table 2 and Figure 5). The AuNPs deposited in the pristine soybean-based fabric showed two low intensity peaks at 87.2 and 83.6 eV, which are assigned to Au4f_5/2_ and Au4f_7/2_ spin-orbit coupling of bulk metallic Au^0^ interacting with the carboxylic groups or the protein amino/peptide functionalities of soybean fibres [54]. This is suggested by the smaller value of binding energy of Au4f_7/2_ in respect of the characteristic parameter for bulk Au (84 eV) [55]. The addition of chitosan to the system clearly shows that the AuNPs were partially oxidized on the fibre surface. Differently to bulk Au, the AuNPs are widely reported to been able to form a weak covalent bond with the amine group of chitosan (NH_2_) and an electrostatic bond between NH_3_^+^ and the negatively charged AuNPs surface [56]. The high-resolution spectrum of the Au4f core level of chitosan/AuNPs coated fabric is characterized by two pairs of peaks related to elemental gold Au^0^ interacting with chitosan (BEs of 83.8 and 87.5 eV) and to the gold oxide state, Au^+^ [57]. The Au^0^ peak is shifted to value nearest to bulk Au providing further evidence that AuNPs were successfully reduced to metallic pure Au bound to chitosan, which act as macromolecular ligand for chelating AuNPs. The presence of the Au^+^ peak due to the unreduced gold on the AuNPs surface, suggests that the AuNPs interaction with chitosan involve several changes in the gold oxidation state, as well as in the chemical state of the reactive sites of chitosan with inter- or intra-molecular coordination [58]. No peaks of Au_3_^+^ at 86 and 90 eV were observed indicating the good chitosan covering of the AuNPs with a full reduction of the AuCl_4_^-^ ions of the initial precursor into Au^0^.

### 3.6. Thermal Properties

Degradation steps associated with temperature rising were identified on the soy fabrics untreated and treated with chitosan and/or functionalized with AuNPs via TGA (Figure 6). The first step of degradation observed between 35 °C and 100 °C refers to the initial volatilization of moisture and solvent from the specimen due to evaporation or dehydration of hydrated cations [59]. This step was more important on the bare soy fabrics, suggesting the presence of more water molecules per repeat unit of PVA and/or bonded with the soybean proteins. On this sample, the second step detected at 295.1 °C (Table 3) can be attributed to the cleavage of the PVA polymeric backbone, which initiates with the degradation of the side chains and progresses to the main chains (at ≈350 °C), culminating in a total mass loss of 89.7% (10.3% of residual mass at 600 °C, Table 3) [60]. Fabrics treated with chitosan experienced the second degradation step at ≈270 °C, which is associated with the initial degradation of the chitosan polymeric chains by means of random chain break and partial deacetylation [61]. Here, the maximum degradation temperature was determined at 324.5 °C (Table 3). At ≈353 °C another degradation step was identified, which could be attributed dehydration and depolymerization reactions of chitosan most likely associated with the decarboxilation of the protonated carboxylic groups [60,62]. In the end (600 °C), only 14.3% of the initial mass remained. Functionalization of the soy fabrics with AuNPs shifted the second degradation step of the fabric by only 3 °C, to ≈292 °C. Negatively charged AuNPs are capable of binding via electrostatic attraction or quasi-covalent bound to positively charged protein molecules. Moreover, the Au^3+^ or Au^+^ layer of unreduced gold on the surface of the Au^0^ nanoparticles can react with the carboxylic moieties onto the fibres surface. However, these interactions are easily broken with the rising in temperature [63]. The absence of a binding agent between the soy fabric and the AuNPs resulted in the formation of large, compact NPs agglomerates widely spaced along the fabric and incapable of contributing to its thermal stability. It is well recognized that AuNPs bind strongly, via covalent bonding, with organic molecules containing amine groups, including chitosan as demonstrated in the XPS analysis. Also, chitosan is known to promote positive capping agent’s electrostatic repulsion between metal NPs, preventing aggregation [36,63]. The TGA profile for the soy fabrics treated with chitosan and functionalized with AuNPs proved just that. The maximum degradation temperature (second step) increased ≈24 °C compared to chitosan treated fabrics, and ≈50 °C compared to the bare and AuNPs coated fabrics (Table 3). Also, the second step of degradation was not as quick to occur (broader curve), suggesting the complexity of interactions formed between the three elements to require more time to break. The last degradation step occurred at 424.1 °C, above all the other tested surfaces, offering definitive proof of the increase in thermal stability of the hybrid fabric with the immobilization of AuNPs. The residue at 600 °C is very similar to the chitosan-coated fabric. Overall, the presence of residues (>9%) after 600 °C heating may be explained by the occurrence of impurities or partial carbonization resultant from uncompleted polymer degradation.

DSC technique measures physical and chemical transformations of a material when subjected to heating. DSC thermograms of the soybean fabrics bare and coated with chitosan and/or AuNPs were acquired between 30 and 350 °C (Figure 7). On all surfaces, the initial endothermic peaks were identified in the temperature range of 76.4 to 80.5 °C (Table 3). These broad peaks are often termed as dehydration peaks and occur due to the evaporation of water molecules bonded to the hydrophilic groups of the polymers or proteins [64]. Because of these peaks wide curve, the peak associated with the denaturation of the soy protein, mainly the 7S (conglycinin) and 11S (glycinin) fractions, that occurs at ≈80 °C was masked [65]. The soybean protein has a molecular weight that varies between 8 and 600 kDa, which can be fractionated into 2S, 7S, 11S, and 15S according to their sedimentation coefficients. The amino acid composition of 7S and 11S, representing about 70 to 80% of the entire protein, determine their major properties [41]. The second peak at ≈230 °C was attributed to the melting temperature (T_m_) of PVA, with an average endothermic energy of 65 J·g^-1^ for all surfaces. It is common to find the T_m_ of PVA at around 200 °C [66]. This shift to a higher temperature may indicate that the blending process has destroyed both the soy and the PVA individual crystalline structures, generating a hybrid fabric based on molecular interactions. For instance, the soy amino acids could potentially interact with the functional hydroxyl groups of the PVA, and consequently increase the T_m_ of the blend [41]. Moreover, at this temperature other fractions of the protein may experience degradation as well [65]. Because of the hybrid network formed, which limits the movement of the PVA and soybean protein segments, complete decomposition of the soy fibres required another degradation step at ≈276 °C. It has been shown that proteins processed in the form of fibres to display an interconnected structure formed at the nanoscale that hinders degradation to higher temperatures (≈300 °C) [67,68]. Shifting of this peak to higher temperatures by chitosan treatment (291.4 °C) confirms this polymer ability to increase the thermal stability of the fabric. Degradation of chitosan chains starts at ≈270 °C as seen by TGA [61]. Loading of AuNPs onto chitosan-treated fabrics improved the overall thermal stability of the hybrid fibres. Still, the enthalpy of the engineered fabric experienced a significant reduction after coating because of the amorphous structure and low crystallinity of chitosan and the AuNPs (33.3 J·g^−1^), indicating that they require lower amounts of energy to undergo decomposition. A fourth broad peak attribute to chitosan is also depicted in the DSC of chitosan and chitosan/AuNPs samples between 320 and 330 °C, which may result from dehydration and depolymerization reactions (data not shown) [62].

### 3.7. Colour Fastness

Two of the most important factors to create durable fabric coatings are the washing fastness and the reflectance properties. The diffuse reflectance spectra of the AuNPs functionalized soy fabrics, pre-treated with chitosan, unwashed and washed (1 to 5 cycles) as well as the bare soybean fabric washed five times are shown in Figure 8. Surface reflectance was determined above 20% for unwashed fabrics, with two important peaks between 420–500 nm and 650–700 nm, respectively blue and red colours. This is to be expected since the fabrics acquired a darker bluish-red colouration after surface modification, as observed in Figure 3b. Fabric’s reflectance increased with the washing cycles, as weakly unbounded chitosan molecules and impurities from the AuNPs functionalisation method were removed. In the end, only strong, chemically bonded NPs remained on the fabric, resisting up to 5 washing cycles (Appendix A). Reflectance was found at ≈34% which was significantly lower of the ≈80% of the washed bare fabric.

The L*, a*, b* and ΔE*, listed in Table 4, were determined for the unwashed and washed chitosan-treated fabrics. The unwashed fabric worked as control. It is clear from the data that the chemical bond generated between the soy fabrics, chitosan and the AuNPs to be chemically strong, resisting the washing cycles. Fabric lightness (L*) was maintained between 62 and 65 throughout the entire test, confirming the stability and homogeneous distribution of the coatings. The chromaticity layer b*, attributed to yellow-blue, and the chromaticity layer a*, attributed to red-green, increased after the first washing cycle moving in the direction of the blue and red, respectively. The ΔE* values before and after washing indicate the changes in colour and can also be considered as a direct measure of the degree of stability of the AuNPs. Little differences were observed between washing cycles, which attest to the high washing durability of the functionalised fabrics.

### 3.8. UV-Light Protection

The values of the UPF and the percentage of UV transmission for UVB (280–315 nm) and UVA (315–400 nm) radiation on bare and coated fabrics are listed in Table 5. Data shows that a higher protection against UV radiation is provided by the AuNPs functionalized fabrics pre-treated with chitosan. Bare soy fabrics were only capable of an average UPF of 7.3, blocking 87.7% of UVB radiation and 78.7% of UVA. These values put the bare fabrics at the bottom of the UPF classification grade, which rates the ability of the fabric to block UV from passing through and reaching the skin, with a factor of +5, considerably below the acceptable “fair to good” rate attributed to +15 [69]. The value with only AuNPs immobilized on the fabric surface reach an UPF of 14.6 indicating a “fair“ +10 rate due to the low amount of NPs able to fix onto the fibres surface. On the other hand, chitosan is able to display and excellent UPF (62.2) rating +50. However, this value reaches a “very good” rating (<50) after few washing cycles. The average transmittance values of the soy fabrics decreased after AuNPs/Chitosan functionalization, demonstrating that the high number of immobilized NPs prominently enhanced the ability of UV-blocking of the soy knitted fabrics. Both transmittance values in UVB and UVA regions decreased with the presence of the AuNPs, blocking a total of 99.2% of UVB and 99.4% of UVA. These values point to the ability of functionalized fabrics to prevent radiation from penetrating both the outer layer of the skin, the epidermidis (UVB), and the middle layer of the skin, the dermis (UVB) [70]. Here, the UPF value was determined at 288.7, ≈40 times fold the UPF of the bare fabrics. Results indicate the AuNPs/Chitosan functionalized fabrics to provide “excellent” UV shielding with a factor of +50 UPF even after several washing cycles. These data are consistent with previous report which, however, was not able to reach the protection ability obtained in this work [5].

### 3.9. Antimicrobial Action

Fabrics’ antimicrobial efficacy was tested against *S. aureus* and *E. coli* at each stage of modification (Table 6), using the standard shake flask method. Both chitosan and AuNPs were successful in fighting bacteria on their own (>95% against *S. aureus* and >52% against *E. coli*). Many reports suggest their bactericidal performance to be a result of the interactions established with other polymers or ionic forms [71,72]. Still, it is undeniable that their presence contributes to the reduction of bacterial action. In this particular case, they were more important against Gram-positive bacteria. The difference in the cell wall structure and composition between Gram-positive and Gram-negative bacteria may explain this discrepancy. The cell wall of Gram-positive bacteria is composed of a thick peptidoglycan layer formed of linear polysaccharide chains cross-linked by short peptides, resulting in a 3D rigid structure. The cell wall of Gram-negative strains is more structurally and chemically complex with a thin peptidoglycan layer adjacent to the cytoplasmic membrane and a lipopolysaccharidic outer membrane. The hydrophilic nature of the outer membrane and the presence of periplasmic-space enzymes, capable of degrading molecules introduced from the outside, on Gram-negative bacteria, explain the difficulty of antimicrobial agents in penetrating the cell wall and, consequently, the bacteria superior antimicrobial resistance [60,73,74].

Data established the soy knitted fabrics treated with chitosan and functionalized with AuNPs as the most efficient in fighting bacteria, with a reduction of ≈99.94% (log_10_ 3.23) of *S. aureus* and ≈96.26% (log_10_ 1.43) of *E. coli*, after 24 h culture. This remarkable improvement can be explained by the introduction of chitosan as a binding agent in between soy and AuNPs. Chitosan is capable of enhancing the surface activity by changing the surface charge and generating new active binding sites, such as NH^+^ groups. In solution, the amino functions of chitosan are protonated and the resultant soluble polysaccharide is positively charged. Its cationic nature gives chitosan the ability to form complexes with a variety of polyanions. The negatively charged AuNPs are then attracted and bind via electrostatic forces to the positively charged chitosan, forming a stable complex [24,75,76]. The ability of AuNPs to target diverse bacterial strains is intimately associated with their well-developed surface chemistry, chemical stability and appropriate small size. It has been reported that biophysical interactions occur between NPs and bacteria through aggregation, biosorption and cellular uptake, causing membrane damage and toxicity [77]. Bacteria cell membranes are somewhat permeable and allow only small molecules to pass through. Thus, small AuNPs with large specific surface area are capable of penetrating the bacteria cell wall more easily than larger-sized NPs. They generate aggregates within the bacteria cell, causing further damage, and ultimately leading to the loss of bacteria activity and to their death. However, due to the immobilized nature of the AuNPs onto the fabric surface the antimicrobial action in this case is more probably related to the AuNPs ability on generating reactive oxygen species, i.e., Au^3+^, which increase the oxidative stress of microbial cells (oxidation of the bacteria molecular structure). AuNPs can then interact with bacteria via their Au^3+^ ions, forming electrostatic interactions with the negative groups of the cell membrane, and this way facilitate cellular uptake, microbial penetration and finally antimicrobial action [13,77].

The control fabric displayed antibacterial activity against both pathogens. This happens because bacteria can get entrapped within the fabric fibres decreasing the amount in solution. Also, it has been reported that the tendency of *S. aureus* to form cellular aggregates in response to surface hydrophilicity, explaining the difference in ≈38% of bacteria reduction between *S. aureus* and *E. coli*. [78].

## 4. Conclusions

The inclusion of AuNPs within a chitosan matrix revealed excellent antimicrobial properties against both Gram-positive and Gram-negative bacteria, UV-light protection (UPF + 50) and washing fastness up to five laundry cycles. Besides the great potential for biomedical applications, the coated AuNPs/chitosan soybean-based fibres displayed improved thermal stability and interesting features for structural colouration of textiles. XPS analysis revealed the complex chemical nature of the interactions among gold species, chitosan and soybean fibres. It was proved that AuNPs were reduced to metallic pure Au bound to chitosan, which act as a macromolecular ligand for chelating AuNPs. Despite the absence of Au^3+^ in the XPS analysis, it was hypothesized that the antimicrobial action of the AuNPs in solution could be driven by the production of oxidized Au species, such as Au^+^ and Au^3+^, leading to the generation of reactive oxygen species. Further investigation will be needed to prove the relationship between AuNPs chemical state and their antimicrobial mode of action.

## Figures and Tables

**Figure 1 nanomaterials-09-01064-f001:**
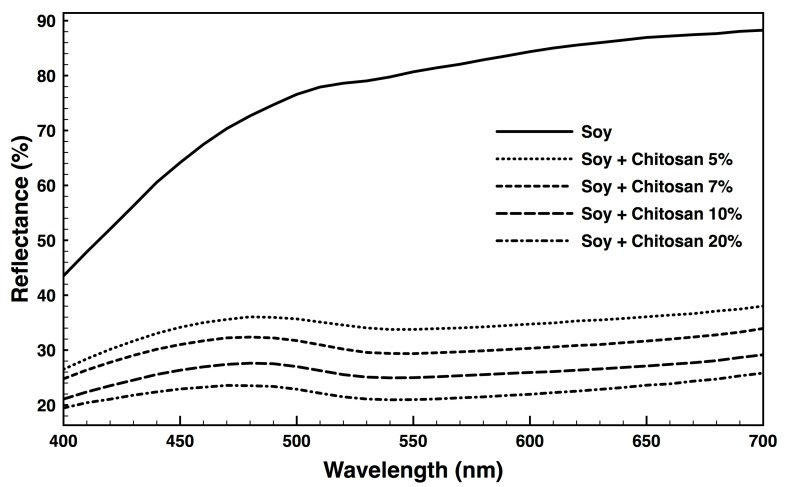
Fabric’s reflectance after 5, 7, 10 and 20% chitosan dilutions surface treatment, obtained in the range of 400 to 700 nm at standard illuminant D65.

**Figure 2 nanomaterials-09-01064-f002:**
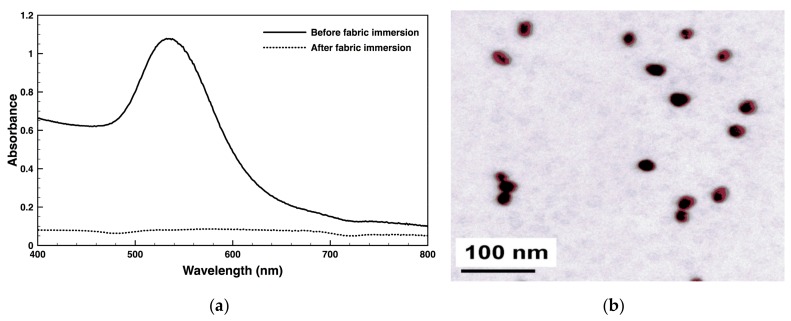
(**a**) Ultraviolet (UV)-visible spectra of Gold nanoparticles (AuNPs) colloidal suspension (400–800 nm). (**b**) Transmission electron microscopy (TEM) micrograph of the relatively monodispersed, highly spherical AuNPs.

**Figure 3 nanomaterials-09-01064-f003:**
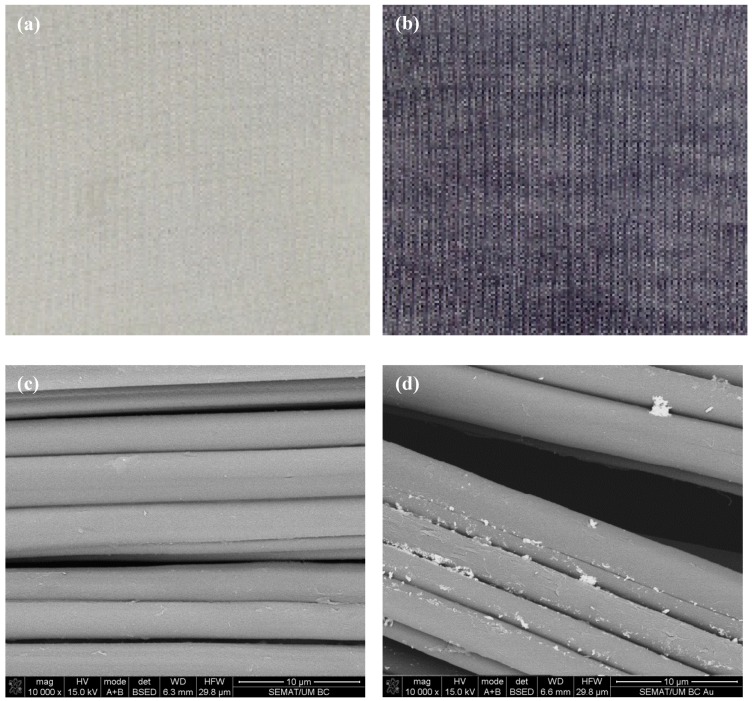
Visual results and scanning electron microscope (SEM) micrographs at 10,000 × magnifications of soybean fabrics (**a**,**c**) bare and (**b**,**d**) functionalised with AuNPs via 20% *v/v* chitosan.

**Figure 4 nanomaterials-09-01064-f004:**
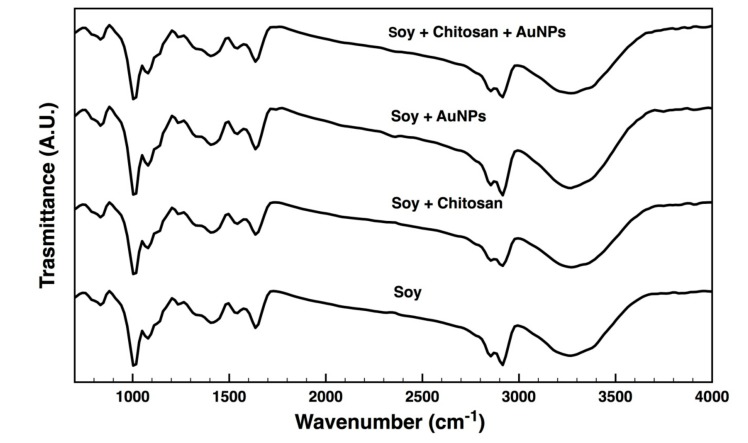
Attenuated Total Reflectance-Fourier-Transform Infrared Spectroscopy (ATR-FTIR) spectra of soy fabrics untreated, treated with 20% chitosan, functionalized with AuNPs, and treated with 20% chitosan and subsequently functionalized with AuNPs (4000–700 cm^−1^).

**Figure 5 nanomaterials-09-01064-f005:**
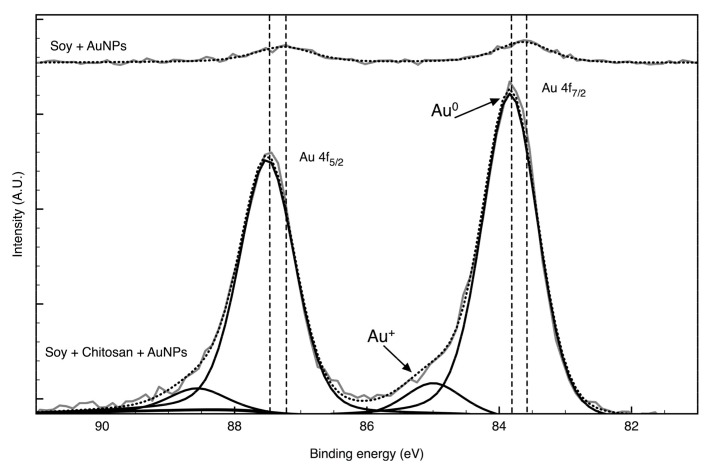
High-resolution deconvoluted XPS spectra with relative areas of the Au4f binding energy region on soybean fibres functionalized with AuNPs before and after chitosan treatment.

**Figure 6 nanomaterials-09-01064-f006:**
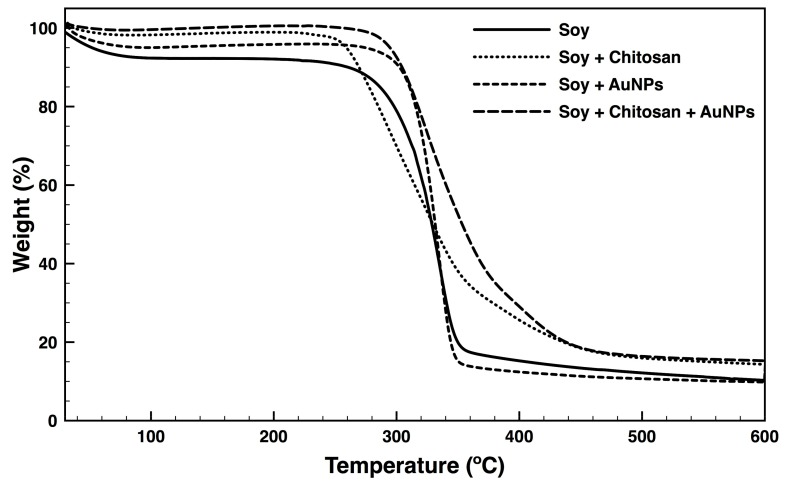
Thermal Gravimetric Analysis (TGA) of soybean fabrics bare and coated with chitosan and/or AuNPs from 30 to 600 °C, performed at a heating rate of 20 °C min^−1^ in a nitrogen atmosphere.

**Figure 7 nanomaterials-09-01064-f007:**
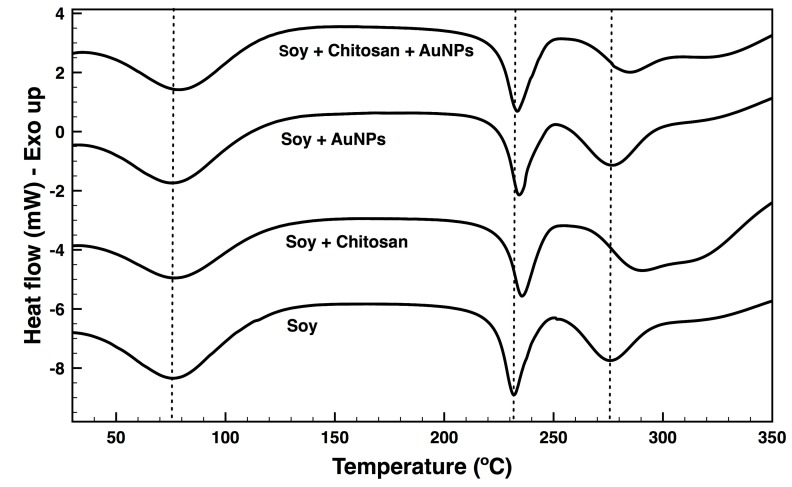
DSC thermograms of soybean fabrics bare and coated with chitosan and/or AuNPs in a temperature range from 30 to 350 °C, performed at a heating rate of 20 °C min^−1^ in a nitrogen atmosphere.

**Figure 8 nanomaterials-09-01064-f008:**
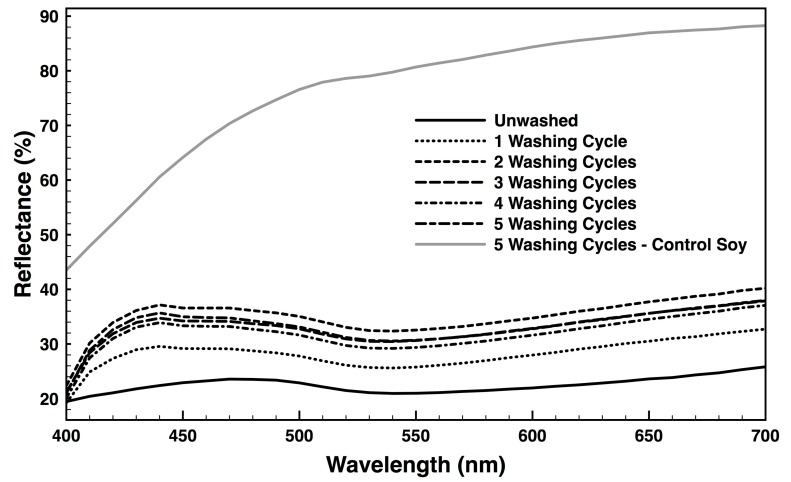
Reflectance of AuNPs functionalized fabrics, pre-treated with chitosan, unwashed and after 1 to 5 washing cycles, obtained in the range of 400 to 700 nm at standard illuminant D65.

**Table 1 nanomaterials-09-01064-t001:** Relative chemical composition and atomic ratio of soybean fabric samples coated with chitosan and gold nanoparticles (AuNPs).

	Chemical Composition (%)						Atomic Ratio	
Samples	C1s	N1s	O1s	Si2p	S2p	Au4f	O/C	N/C
Soy	86.66	1.59	9.58	1.98	0.19	-	0.11	0.02
Soy + Chitosan	72.55	3.63	23.15	0.55	0.12	-	0.32	0.05
Soy + AuNPs	80.58	2.26	14.63	2.34	0.14	0.05	0.18	0.03
Soy + Chitosan + AuNPs	70.53	4.34	23.85	0.30	0.15	0.74	0.34	0.06

**Table 2 nanomaterials-09-01064-t002:** Results of the deconvolution analysis of the C1s, N1s, O1s and Au4f peaks of soy fabrics coated with chitosan and AuNPs. Reported binding energies have an associated error of ± 0.1 eV.

Relative Area Corresponding to Different Chemical Bonds (%)					
	BE (eV)	Soy	Soy + Chitosan	Soy + AuNPs	Soy + Chitosan + AuNPs
C1s	285.0	66.0	49.0	74.6	48.5
	287.1	5.7	-	-	-
	287.9	-	11.2	4.9	10.9
	285.9	25.5	-	-	-
	286.5	-	36.7	17.9	38.1
	288.5	2.8	-	2.6	-
	288.9	-	2.8	-	2.5
O1s	534.5	6.6	4.6	-	2.6
	532.3	93.4	-	100	-
	532.8	-	84.7	-	-
	531.3	-	10.7	-	-
	532.9	-	-	-	85.2
	531.5	-	-	-	12.2
N1s	400.0	100.0	100.0	100.0	84.8
	402.0	-	-	-	15.2
Au4f	87.2	-	-	44.4	-
	87.5	-	-	-	40.1
	83.6	-	-	55.6	-
	83.8	-	-	-	51.4
	85.0	-	-	-	5.1
	88.6	-	-	-	3.5

**Table 3 nanomaterials-09-01064-t003:** Main Differential Scanning Calorimeter (DSC) thermal transitions, Thermal Gravimetric Analysis (TGA) weight loss temperature peaks and residual weight (*n* = 3; S.D. < 1%).

Samples	Tm (°C)	ΔH (J g^−1^)	T Peaks of 1st Derivative (°C)	Residue (%) at 600 °C
Soy	76.4; 231.6; 275.8;	159.4; 65.5; 58.8	295.1	10.3
Soy + Chitosan	78.0; 235.4; 291.4	127.5; 65.0; 90.6	324.5; 352.9	14.3
Soy + AuNPs	77.1; 234.0; 277.0	130.5; 53.4; 61.7	292.3	9.8
Soy + Chitosan + AuNPs	80.5; 233.2; 284.2	312.5; 70.7; 33.3	348.3; 424.1	15.3

**Table 4 nanomaterials-09-01064-t004:** Overall colour difference (ΔE*) and shift of the coordinates of the colour in the cylindrical colour space, lightness (L*), red-green (a*), and yellow-blue (b*), of AuNPs functionalized fabrics, pre-treated with chitosan, before and after washing cycles (unwashed, 1, 2, 3, 4 and 5).

	Unwashed	1	2	3	4	5
L*	65.42	62.83	64.93	63.29	62.29	64.67
a *	1.80	2.86	2.80	2.95	3.15	2.83
b *	2.63	−2.87	−2.76	−2.44	−2.91	−2.82
ΔE *	-	6.17	5.50	5.60	6.50	5.59

**Table 5 nanomaterials-09-01064-t005:** Ultraviolet protection factor (UPF) and percent of UV-light transmission (UVB radiation, 280–315 nm, and UVA radiation, 315–400 nm) of bare, chitosan, AuNPs and AuNPs functionalized soy fabrics pre-treated with chitosan.

Samples	UPF ± SD	UPF Classification	Average UVB Transmittance (%)	Average UVA Transmittance (%)
Soy	7.3 ± 3.8	+5	12.3	21.3
Soy + AuNPs	14.6 ± 2.4	+10	6.2	8.2
Soy + Chitosan	62.2 ± 2.3	+50	1.1	3.8
Soy + Chitosan + AuNPs	288.7 ± 0.1	+50	0.2	0.5

**Table 6 nanomaterials-09-01064-t006:** Fabrics’ antimicrobial efficacy against *S. aureus* and *E. coli* assessed quantitatively following the standard shake flask method (ASTM-E2149-01) for a 24 h incubation period. Results are present in the form of mean % bacterial reduction ± SD and log_10_.

Samples	*S. Aureus*		*E. Coli*	
	% ± SD	log_10_	% ± SD	log_10_
**Soy**	51.79 ± 4.17	0.32	13.82 ± 8.27	0.06
**Soy+Chitosan**	97.02 ± 0.88	1.53	52.85 ± 9.86	0.33
**Soy+AuNPs**	98.70 ± 0.23	1.89	69.43 ± 10.7	0.51
**Soy+Chitosan+AuNPs**	99.94 ± 0.02	3.23	96.26 ± 0.51	1.43

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
