# Peer review of "Multifunctional Chitosan/Gold Nanoparticles Coatings for Biomedical Textiles"

_nanomaterials, 2019, doi:10.3390/nano9081064_

Reviewer 1 Report

This is a nice, clear and interesting work, meriting publication.  

Some minor remarks

i) Literature is too extensive. 

ii) Some more details on ALT-B TOUCH 35 equipment would be useful.

Author Response

General Observations:

This is a nice, clear and interesting work, meriting publication.

R. We appreciate the Reviewer's comment and thank him for his insights.

Specific Comments:

1) Literature is too extensive.

R. As requested, we have reduced as much as possible the literature. However, due to the detailed work done and the many experimental tests conducted more investigation was required to explain and support the acquired data, therefore increasing the number of citations.

2) Some more details on ALT-B TOUCH 35 equipment would be useful.

R. Additional information concerning this equipment was added to section 2.3 in the manuscript.

 Reviewer 2 Report

The manuscript by Zille and coworkers reports a comprehensive study on the properties of soybean fabric coated with chitosan and with the inclusion of AuNPs. The work is very well written and the conclusions are sound. Therefore, I recommend publication of the manuscript after the few issues reported here below are addressed. Main concerns regard the variety of possible explanations the authors offer for the AuNPs-chitosan interaction and better performance as respect to chitosan or AuNPs loaded fabrics. Those explanations should be better harmonized.

- p.2 2nd paragraph: "There have been..." Please cite at least a review.

- Please, cite and discuss the results described in Biomacromolecules (2017), 18, 11, 3766-3775.

- p.7 bottom: "Its primary amines can be chemically protonated, serving as chelation sites for adsorption and binding of a number of metal cations". Please, explain this statement, also in view of other explanations reported in the literature, such as the one proposed for AgNP-cellulose interaction in [Ye, D., Zhong, Z., Xu, H. et al. Cellulose (2016) 23: 749. https://doi.org/10.1007/s10570-015-0851-4]. I would say that no free metal cations are present once the AuNPs are formed, but perhaps I'm missing something. Moreover, electrostatic repulsions between cations (protonated amines and metal cations) should hamper the interaction. It seems AuNPs adhesion would be helped by neutral primary amines rather than protonated ones. Can the authors please comment on that point? In this connection, can the authors please explain their statement reported in p. 10: "The presence of coordinated (?) protonated amino groups confirms the predominance of amine sites in chitosan for metal sorption charge transfer from amine sites to AuNPs"? Besides, how does the "chelation" effect of protonated amines (p.7) meet with the abovementioned "charge transfer"? A comment on how the statement reported in p.7 (top): "It also suggests that the relative stability of the AuNPs in the dispersion to result from the electric repulsion of the negative charge of the citrate ions adsorbed onto the NPs surface" can be reconciled with the rest of the discussion would be also appreciated.

- p.7 top: the size of AuNPs is rather big. Did the authors select this particle size for any particular reason? In the antibacterial results section, the authors state that a small particle size should be preferred. Could the authors please comment on that?

Author Response

General Observations:

The manuscript by Zille and coworkers reports a comprehensive study on the properties of soybean fabric coated with chitosan and with the inclusion of AuNPs. The work is very well written and the conclusions are sound. Therefore, I recommend publication of the manuscript after the few issues reported here below are addressed. Main concerns regard the variety of possible explanations the authors offer for the AuNPs-chitosan interaction and better performance as respect to chitosan or AuNPs loaded fabrics. Those explanations should be better harmonized.

 R. We appreciate the Reviewer's comments and will try to address all the concerns highlighted.

 Specific Comments:

1) p.2 2nd paragraph: "There have been..." Please cite at least a review.

 R. A review was added as requested.

 2) Please, cite and discuss the results described in Biomacromolecules (2017), 18, 11, 3766-3775.

 R. Even though the results exposed in this paper are very interesting, the fact the authors used silver (Ag) inlaid AuNPs alters completely the performance of the NPs being no longer possible to compare the two researches. Ag is by itself antimicrobial; therefore, the use of chitosan combined with Ag-Au NPs in biomedical applications cannot be looked in a similar way as our soy fabrics loaded with AuNPs. Looking more in detail the 2017 publication, the authors did not observe any antimicrobial activity on the samples with only gold or gold/chitosan, probably due to the low concentration or to inadequate antimicrobial test (as reported in the introduction, the inhibition zone method presents several drawbacks). Moreover, no deconvolution of the XPS high-resolution spectra of the O1s, C1s and N1s envelopes were performed for all the samples limiting the data interpretation especially in the interaction between gold and chitosan.

 3) p.7 bottom: "Its primary amines can be chemically protonated, serving as chelation sites for adsorption and binding of a number of metal cations". Please, explain this statement, also in view of other explanations reported in the literature, such as the one proposed for AgNP-cellulose interaction in [Ye, D., Zhong, Z., Xu, H. et al. Cellulose (2016) 23: 749. https://doi.org/10.1007/s10570-015-0851-4]. I would say that no free metal cations are present once the AuNPs are formed, but perhaps I'm missing something. Moreover, electrostatic repulsions between cations (protonated amines and metal cations) should hamper the interaction. It seems AuNPs adhesion would be helped by neutral primary amines rather than protonated ones. Can the authors please comment on that point?

 R. It is well documented that the citrate reduced gold nanoparticles are negatively charged. In fact, citrate ions are attracted towards the surface of the AuNPs, conferring some degree of stability to the NPs and a negative charge. On their own AuNPs are very unstable in colloidal suspensions [Langmuir 1995,11, 3712-3720]. In our case the same is observed. As such, the citrate ions surrounding the AuNPs are then available to interact with the amine and hydroxyl groups from chitosan and generate a strong bond. Moreover, on one hand, the deconvolution of the C1s and O1s high-resolution spectra demonstrated the absence of interaction of the AuNPs with the oxygen moieties of the fabric or chitosan and, on the other hand, the deconvolution of the N1s envelope showed the presence of coordinated protonated amino groups (as a new peak at 402 eV) only in the Au/chitosan samples, confirming the predominance of amine sites in chitosan for metal sorption due to the charge transfer from amine sites to AuNPs. Finally, the XPS of the Au interacting with chitosan clearly showed the presence of oxidized gold in the form of Au+ but not Au3+ indicating that the charge comes from the unreduced gold on the AuNPs surface and not from the unreacted AuCl4- ions. We have added these details to the discussion, sections 3.2 and 3.3, to clarify our statements.

 4) In this connection, can the authors please explain their statement reported in p. 10: "The presence of coordinated (?) protonated amino groups confirm the predominance of amine sites in chitosan for metal sorption charge transfer from amine sites to AuNPs"? Besides, how does the "chelation" effect of protonated amines (p.7) meet with the abovementioned "charge transfer"? A comment on how the statement reported in p.7 (top): "It also suggests that the relative stability of the AuNPs in the dispersion to result from the electric repulsion of the negative charge of the citrate ions adsorbed onto the NPs surface" can be reconciled with the rest of the discussion would be also appreciated.

 R. The electrostatic attraction between positively charged amino groups of the chitosan chains and the negatively charged AuNPs results in nanoparticles stabilization and coordination of the protonated amino group of chitosan on the nanoparticle surface, due to the charge transfer on nitrogen, resulting in strong bonds (DOI: 10.1021/am508094e). Moreover, the peak at 402 eV for N1s that only appeared in the sample with both AuNPs and chitosan confirms that the distribution of the AuNPs/chitosan interactions is not the same on the surface and in the bulk of the nanoparticles (DOI: 10.1016/S0927-7757(00)00678-6). We have added these details to the discussion in section 3.5, to clarify our statements.

 5) p.7 top: the size of AuNPs is rather big. Did the authors select this particle size for any particular reason? In the antibacterial results section, the authors state that a small particle size should be preferred. Could the authors please comment on that?

 R. In fact, what is stated is that only in cases of leaching a small particle size is preferred. In this particular case, because the binding of the Au NPs to the soy fabric is strong and, thus, no leaching occurs, the size of the NPs is not relevant. The antimicrobial activity is an effect of reactive oxygen species formed by the Au NPs, that then interact with the bacteria cell membrane, compromising its integrity.